# Potential facilitators and inhibitors to the implementation and sustainability of the community-based tuberculosis care interventions. A case study from Moshupa, Botswana

Gabalape Arnold Sejie[1,2]*, Ozayr H. Mahomed[1,3]

1 Discipline of Public Health Medicine, University of KwaZulu, Natal, Durban, South Africa, 2 Department of Health Promotion and Education, Boitekanelo College, Gaborone, Botswana, 3 Dasman Diabetes Institute, Kuwait City, Kuwait

* sejiearnold@yahoo.com

## Abstract

### Background

Eliminating Tuberculosis is one of the targets of Sustainable Development Goal Three. Decentralizing TB care beyond health facilities by leveraging community involvement is crucial for safeguarding effective tuberculosis care. In this study, we explored potential facilitators and inhibitors of the implementation and sustainability of community-based interventions for the prevention and treatment of TB in the Moshupa district, Botswana.

### Methods

This study adopted a qualitative approach using a collective case design. An interpretive paradigm based on relativist ontology and subjectivist epistemology along with abductive research logic was used. The study enrolled treatment supporters of tuberculosis patients diagnosed with drug-susceptible tuberculosis between January 2019 and December 2019 in Moshupa Village for semi-structured interviews, Health care professionals for in-depth interviews, and e community leaders for focus group discussions. Clinic-based observations in Mma-Seetsele clinic were also conducted to corroborate the participants' views. The data collected was analyzed using the NVivo version 12 software package, and statements of the participants were presented as quotes to substantiate the issues discussed.

### Results

This study highlighted effective partnerships between health services and external stakeholders, community empowerment, and the availability of policies and standard operating procedures as facilitators of community TB implementation and sustainability. However, Insufficient funding, low service provider training, policies not embracing age and educational eligibility for treatment supporters, shortage of equipment, medicines, and supplies,

**Funding:** The author(s) received no specific funding for this work.

**Competing interests:** The authors have declared that no competing interests exist.

**Abbreviations: CHW**, Community health worker; **DOTS**, Directly Observed Treatment Short Course; **IRB**, Institutional Review Board; **NGO**, Non-Governmental Organization; **PTB**, Pulmonary TB; **TB**, Tuberculosis; **WHO**, World Health Organization.

inadequate transport availability and incentives to meet clients' basic needs, paper-based systems, inadequate supervision, incomplete data reporting, and low service quality affected the Community TB program efficacy and sustainability in Moshupa. We also found that there was low service provider motivation and retention and that clients had low trust in treatment supporters.

## Conclusion

The findings of this study imply that the operational effectiveness of the community TB care approach to disease elimination is compromised; therefore, initiatives addressing the key components, including the availability of resources, governance arrangements and supportive systems for community health workers, are required for successful community TB implementation and sustainability.

## Introduction

Tuberculosis [TB] remains a serious public health threat to people of all sexes and ages, particularly in Botswana. In 2022, the World Health Organization (WHO) projections indicated TB as one of the leading causes of death worldwide [1]. Globally, an estimated 10.6 million people fell ill with TB, of which 1.6 million died from the disease, including 187 000 deaths among HIV-positive people in 2021 [2]. The burden of disease varied enormously among countries, with a global average of approximately 137 cases per 100 000 population [2]. The WHO African region had the highest TB burden, with TB incidence of 237 cases per 100 000 population and accounting for 23% of the global TB burden [2]. The proportion of TB cases co-infected with HIV was highest in countries in the WHO African Region, exceeding 50% in parts of southern Africa [2].

Botswana carries a notable proportion of the global TB/HIV burden and is listed among the 30 high-burden country lists for TB/HIV [3], with an incidence rate of 235/100,000, TB-related death rate of 35/100 000 among the HIV negative and 62/100 000 among the HIV positive, and a treatment coverage of 45% in 2020 [4]. The overall treatment success rate for the new and relapse cases treated in 2019 for Botswana was 72%, with 73% and 70% among the HIV coinfected and multi drug resistant cases, respectively [4]. Moshupa, a rural village in Botswana, had the highest TB notification rate in the country with 754/100 00 cases, and a 57% treatment success rate in 2017 [5].

The WHO in its resolve to intensify the fight against TB adopted the End TB Strategy, which envisions a TB-free world by 2035 [3]. This strategy emphasizes the importance of taking into consideration the journey of a TB patient through a series of interlinked settings and facilities, one of which is to decentralize TB care beyond health facilities and harness the contribution of communities through the provision of effective community-based directly observed therapy (DOT) to TB patients at greatest socio-economic risk, thus improving and strengthening community mobilization and treatment support [5].

The Botswana National Tuberculosis and Leprosy Program (BNTP) adopted community TB care (CTBC) as recommended by the WHO since 2004 to increase community involvement in tuberculosis activities [6]. In this model, all patients start anti-tuberculosis Therapy (ATT) at the facility level for at least two weeks. At the end of this period, patients who choose community TB Care (CTBC) are transferred to a DOT point of their choice [6, 7]. These DOT points can be household, workplace, school, church or any place where DOT can be provided.

The DOT points are operated by a community health worker (CHW) or treatment supporter. The choice of DOT point depends on the mobility of the patient however patients who are not mobile receive DOT at home from an identified treatment supporter. Patients remain registered at the health facility while they are on CTBC and reviewed at the health facility monthly and when required [6, 7]. The TB treatment supporter is someone identified in the community by the patient whom they trust and want to have as a supervisor of their ATT. A relative, spouse, neighbour, friend. The TB focal person educate the treatment supporter about TB and their role in CBTC [6, 7]. The roles and responsibilities of the treatment supporters involves, collection of treatment from the DOT Point every 2 weeks, keeping drug supplies, delivering DOT on daily basis to the TB patient, recording adherence on TB Patient Card, provision of psychological support to patients, monitoring drug side effects and reports any problems to the TB focal person immediately and ensures sputum collection and transport to the facility. However, despite the proven effectiveness of these low-cost strategies in controlling TB, Moshupa still experiences high TB incidence rates with low successful treatment outcomes compared to the national and WHO target, thus posing a grave danger by encouraging the development of drug-resistant strains, one of the greatest threats to TB control [5]. Given this situation, there is an urgent need to review community TB programmes. Understanding stakeholder and service provider experiences in community TB care will not only contribute to enhancing the process but will also inform policy directions for better implementation to achieve the program's targets and goals. Thus, the aim of this study was to explore factors supporting and hindering the implementation and sustainability of community-based interventions for the prevention and treatment of TB in Moshupa village, Botswana.

## Conceptual framework

We adopted a conceptual framework that identifies critical areas for measuring the performance of community health worker (CHW) programs within community health systems. Although community health systems are inherently nonlinear and complex, the framework structure uses the common input-process-output-outcome logic model approach and has four areas: inputs, programmatic processes, community health performance outputs, and outcomes (**S1 Fig in** S1 File) [8]. Specific measurement domains and subdomains are defined under each of these categories, with operational definitions in S1 Table.

## Materials and methods

### Setting

The study was conducted in Moshupa Village. The village is situated in the Southern District of Botswana, with a population of 23,858 as per the 2022 census [9]. The village lies between latitude 24.7763˚ South and longitude 25.4141˚ east. The people of Moshupa are the Bakgatla-ba-ga Mmanaana [10, 11]. There are two clinics and one health post serving the entire village. Patients with TB receive free treatment (both intensive and continuous phases) in local clinics. The community was chosen due to its poor performance metrics, with one of the lowest treatment success rate in the country in 2017 [11].

### Study design and population

This study adopted a qualitative approach using a collective/multiple case design. An interpretive paradigm based on relativist ontology and subjectivist epistemology along with abductive research logic was used. The study population consisted of TB treatment supporters, village community leaders, District Health Management Team workforce (head of DHMT, head of

preventative, TB coordinator and focal persons, facility heads, nurses, non-governmental, and international organizations offering TB services in the study area (Botswana University of Maryland Medical School Health Initiative (BUMMHI) Community health care worker, and data support officer). The study was conducted between February and October 2022.

## Procedures

The participants were selected using a purposive sampling technique. This technique was chosen to reach persons who had experienced the phenomenon of this study, TB treatment, care, and control. Ten (10) health care professionals, including the District Management Team (DHMT) head, head of preventative, district TB coordinator, facility heads, nurses, Botswana University of Maryland Medical School Health Initiative (BUMMHI) TB screener, and data support officers, were enrolled in the study and provided with the opportunity to participate in in-depth interviews (IDI), this was chosen to gather information on experiences with program functioning from administrative, organizational, and/or individual perspectives. Twenty-two (22) TB treatment supporters participated in semi-structured interviews (SSI), this category was chosen due to participants perspectives, varying experiences, attitudes, and motives when caring for TB patients (SSI). Eight (8) community/opinion leaders, including community elders and tribal herdsmen, were also chosen to gather data on the involvement of community stakeholders in program implementation, the availability of multiple voices in planning, implementing, and decision-making consultation, community engagement, and cultural appropriateness through a single focused group discussion (FGD) (conducted by the lead author). To support the participants' opinions, clinic-based observations using a standard tool was undertaken at one high- volume health facility (Mma-Seetsele clinic) (conducted by the lead author). The sample size for this study, including the number of FGDs, SSIs, IDIs, and participants, was considered adequate based on the theoretical saturation, whereby no novel findings emerged from the subsequent interviews [12]. Data was collected using researcher developed tools and guides for each participant group. Appointments were booked with all participants to ensure that the interviews took place at their chosen and or convenient time and sites. Interviews were conducted in either English (for all health professionals) or Setswana (for treatment supporters and community leaders). All data collection was done by Principal investigator G.A.S alongside a trained research assistant, joint (Researchers) assessment of the sampling approach and ongoing review of transcripts to explore areas for further probing was ensured to maximize the quality of the data collected.

## Data analysis

Audio recordings of focus group discussions together with field notes from in-depth, semi-structured interviews and observations were transcribed and translated into the English language. During the course of empirical material collection, transcription was performed regularly, managed, and stored in the NVivo version 12 software. Initial transcriptions were followed by cross-checking with field notes developed by the researchers during the interview stage. The four-step approach: prepare, explore, specification, and integration (PESI) for the empirical material interpretation process was used to address the empirical outcome to seek an explanation of the research questions. The cooperative research process [13] was employed to verify the interpretation of the material and the generation of the framework of value co-creation. The interpretation of empirical material by researchers was presented to the participants for their feedback. After this feedback, the transcribed texts were coded, and concepts were developed. These concepts were then combined to develop the categories. These categories and interpretation results were triangulated using the observation field notes.

### Inclusivity in global research

Additional information regarding ethical, cultural, and scientific considerations specific to inclusivity in global research is included in the S1 File.

### Ethical approval

The study obtained ethical approval from the Biomedical Research and Ethics Committee of the University of KwaZulu-Natal in South Africa dated March 03, 2022 (Protocol reference number: BREC/00003225/2021) and the Botswana Health Research and Development Division dated December 16, 2021 (reference number: HPDME 13/18/1) before the study was conducted. Permission was obtained from the Southern District Health Management Team administration (dated February 17, 2022). Written informed consent was obtained from each participant prior to interviews and discussions.

### Informed consent

Study participants who were able to read were asked to read and sign the consent form and those who could not read and write were allowed to bring someone they comfortable within their family or close friend to read and explain the consent form for them while the investigator documented the process. The consent form described in detail the purpose of the study and provided contact information of the researcher, supervisor and the research ethics committees who granted ethical clearances. Anonymity and confidentiality of the information collected were assured at all times.

## Results

### Sociodemographic attributes of the respondents

Participants in the study totaled forty, including twenty-two supporters of TB treatment, eight community leaders, ten members of the DHMT and stakeholders/partner's clinical staff. There were twenty female and eleven male participants. Twelve of the participants were between the ages of 41 and 50, followed by nine people in the 51–60 age range and one person over the age of 60. Ten of the participants had fewer than three years of experience, whereas 30 had more than four years of professional experience in the field of TB care. In terms of educational background, 26 individuals had completed Primary and secondary school, 11 had completed tertiary, and 3 had no formal education. (S2 Table).

### Qualitative findings

This section presents the key findings from the analysis of barriers and facilitators of community TB program implementation in the Moshupa district health system. The themes were organized in accordance with a framework for evaluating the performance of CHW in primary healthcare systems. The final code list used as the basis for the structured data analysis is shown in S3 Table.

**Facilitators to community TB program implementation and sustainability.** *Input performance measure.* **Policies and standard operating procedures**. This study has noted the existence and availability of policies and Standard Operating Procedures (SOP's) with clear job descriptions, monitoring, and evaluation components. These documents supported the implementation of community TB because there were rules and regulations guiding service delivery.

The following observation was made.

*"The facilities had national TB program policies and SOP's with description of the role and tasks to be performed by community health workers, treatment supporters and health systems perspectives were clearly documented,"*

*–(TB clinic Observation 14th April 2022)*

This was further corroborated by one of the interviewees, however she felt the program's implementation may be hampered by policy omission of age and educational background criterion to determine eligibility of treatment supporters. The following is her recorded opinion.

*"The SOP's and policy are good but do not capture age and educational requirements for eligibility to be a treatment supporter, a younger treatment supporter may seem awkwardness to a patient posing challenge to DOT supervision whereas educational level governs the ability for one to read drugs and dosages thus influencing DOT and documentation"*

*- (IDI, Nurse, Moshupa health post)*

***External stakeholders/partnerships***. The Moshupa District Management Team (DHMT) collaborated with the United States (U.S.) government through the PEPFAR-funded Botswana University of Maryland School of Medicine Health Initiative (Bummhi), The alliance offers technical support, which contributes to a decrease in TB-related morbidity and mortality.
A health worker asserted:

*"The Moshupa village has partnered with (Bummhi) in provision of technical support including patient follow-up. The partnership partially counterbalances TB staff inadequacy in the district, which reduces TB-related morbidity and mortality".*

*- (IDI, TB Coordinator).*

Similarly, another health worker opined that the collaboration facilitates program implementation, as some key metrics have improved over the years since their inception. The following quote reflect her opinion.

*"Even though epidemic control has not yet been achieved, key measurements, such as numbers of people developing TB and numbers of people with TB-HIV co-infection have improved over the years".*

*–(IDI, TB focal person)*

*Performance outputs*. **Empowerment**. The findings reflect that, successful implementation of community-centered TB care depended heavily on community empowerment. Most respondents asserted that community leadership was active, and based on their observations, there were no documented unfavorable opinions of the services offered by CHWs.
A community leader stated that;

*"The community was consulted, I remember I was in village development committees (VDC) then when the health education assistants were moving around the community taking care of patients, and the program has no conflict with the community sociocultural practices or believes".*

*- (FGD with Community leader)*

Similarly, another community leader opined that:

*"The community was consulted, I was one of those who were in the red cross, we would take medications on behalf of patients who were not able to carry out their daily activities, and deliver them at their homes"*

*- (FGD with Community leader)*

A health care worker concurred that stakeholders are engaged in implementation, which led to one health care worker admitting that;

*"The target population and community leaders are engaged in implementation, but more can still be done by involving them in development of the framework or implementation model"*

*- (IDI, TB coordinator)*

**Barriers to community TB program implementation and sustainability.** *Input.* **Program funding**. Participants underscored that the program's objectives were frequently unattainable because of unsustainable TB funding, which has been cited as a challenge to program execution.

A health worker opined that;

*"The DHMT relies on government budget to provide for TB services which is often inadequate to fulfill the program activities thus making it difficult to design the program long-term maintenance and monitoring plans with confidence."*

*–(IDI, TB coordinator)*

This was corroborated by another health worker who felt lack of funding is an obstacle to TB elimination.

*"The programme needs more support but there is a challenge of lack of funds to procure all resources needed."*

*- (IDI, Medical officer)*

***Management information systems***. The lack of electronic community-based information management systems was viewed as a barrier for CHWs in recording home visits, submitting visit-related data to the health system, and connecting it to an evaluation of their performance. Inconsistencies in stored data, largely due to the use of paper-based tools that potentially compromise the security and privacy of the data, have been reported. An interviewee points out that;

*"We do not have any framework of software that facilitates the collection, storage, organization, and distribution of information. Instead, we use patient cards and file them as hard copies, report visit related data is given subjectively to the focal person or coordinator for action."*

*–(SSI, Treatment supporter, Mma-Seetsele clinic)*

This was further substantiated by an observation that;

*"Patients' information was stored in paper files with some prints no longer visible, some torn"*

*- (TB Registers Observation 13ᵗʰ April 2022)*

**Logistics**. Interviewees emphasized the importance of logistics in community TB care and the availability of transport arrangements for CHWs and commodities (equipment, medicines, and supplies). We grouped these factors into two subcategories, as illustrated below.

**Transport**. Participants expressed that transport was mostly unavailable for community health workers (CHW) to physically access the target population or provisions, either in the form of monetary re-imbursement (fare for buses) or physical (bicycle) for treatment supporters to access health facilities, thus a barrier to TB service delivery.

An interviewee asserted;

*"We do encounter transport issues in our line of duty trying to assist our patients, only if we could be assisted with some transport money to board a taxi to collect patient medications and transport them for their reviews would help."*

*–(SSI, Treatment supporter, Moshupa SDA)*

Similarly, another interviewee stated that;

*"Sometimes resources are limited for example lack transport to access TB patients at their homesteads thus impacting negatively on patient management, maybe if our government can consider reimbursement to treatment supporters so that whenever we don't have transport, they can board taxis to bring the patient to health facility as per necessity."*

*–(IDI, Nurse, Moshupa health post)*

**Commodities (equipment, medicines, and supplies)**. Respondents reported often being disappointed by the lack of required commodities in equipment, medicines, and supplies to deliver TB services or support the quality of services. Underlining the need for the availability of commodities to facilitate TB program implementation. A participant asserted that;

*"There is shortage of sputum induction machines, shortage of medications and there is also shortage of materials e.g, files and cards, thus we lose patient information."*

*–(IDI, TB screener, BUMMHI)*

Similarly, another health worker opined that:

*"Sometimes resources are limited, for example TB cards are often not enough, If we can ensure availability of such commodities we can be able to improve TB management."*

*–(IDI, Nurse, Mma-Seetsele clinic)*

This concurred with a community leader who commented that:

*"We do encounter a shortage of medication at times and some tests are often omitted because of lack of machines to conduct those tests."*

*- (FGD with Community leader)*

*Programmatic processes*. We identified two main themes: supportive systems(supervision), and community health worker development (training and incentives). These categories are further illustrated as follows.

***Supportive systems (Supervision)***. Participants expressed a lack of consistent and continued support for problem-solving service delivery and skill development from the program coordinators, primarily due to staffing shortages. This has emerged as a barrier to program implementation because of its potential to affect community health workers' motivation for their work. This is substantiated by the words of the participant pointing out that;

*"There is one TB focal person who also works on shift in outpatient department, when she is not there the program services suffer including supervision of community health worker as there will be no one to oversee the program with other unit staff members engaged with their assignments."*

*- (IDI, Data support officer, BUMMHI)*

In addition, this was corroborated by another interviewee who opined that;

*"In our facility, workers are willing to be involved in TB care but there is no support/supervision from DHMT and TB coordinator, this lack of supervision leads to poor morale thus hesitancy to give their all in TB and its management."*

*–(IDI, Nurse, Moshupa health Post)*

***Community health worker development***. Interviewees emphasized the importance of community health worker development in community TB care, recruitment, training, and incentives. We grouped these factors into three sub-categories.

***Recruitment***. Participants expressed dissatisfaction with the method of identifying, selecting, and onboarding community health workers, citing random selection as opposed to their desire to work in TB programs. This has been identified as impeding program success because workers lose their motivation.

An interviewee asserted:

*"Facility staff recruitment in TB care, eg TB focal person is not based on interest but rather unit task allocation potentially leading to unmotivated TB care employees who in turn do very little in delivery of TB services thus impacting the program negatively."*

*–(IDI, Nurse, Moshupa health post)*

Similarly, one health worker expressed her dissatisfaction citing that;

*"Treatment supporter selection is based on patient preferences, no consideration of other engagements such as working, this often leads to supporters leaving patients with medication to take on their own."*

*–(SSI, Treatment supporter, Moshupa health post)*

***Training***. Inadequate TB training was portrayed as an important challenge faced by the programme. For example, a few participants reported a lack of trained personnel to deliver TB care, mostly due to lack of funds and failure of those trained to orientate new staff, thus challenging program implementation.

A health worker stated that;

*"The TB training is limited due to funds unavailability and staff that has been trained not orientating the new staff. This hampers TB progress to national strategy targets"*

*–(IDI, TB coordinator)*

This was substantiated by another interviewee who opined that;

*"I was never trained nor given proper coaching concerning the supervision of TB patients, I just self-taught myself on how a TB patient is taken care of."*

*–(SSI, Treatment supporter, Moshupa SDA)*

**Incentives**. Community TB implementation has been hindered by a lack of CHW incentives. Participants emphasized that a lack of monetary and non-monetary rewards tied to performance could have an impact on provider motivation and performance.
A treatment supporter emphasized that:

*"My wish is that we can be assisted with food because the socio-economic status is often low, and we don't have money since we not working anywhere."*

*–(SSI, Treatment supporter, Mma- Seetsele clinic)*

Although relying on unpaid volunteers appears to make community DOT cheaper, its effectiveness is threatened by volunteerism, as an interviewee points out:

*"There is lack of incentives for treatment supporters hence it becomes difficult to enroll patients who stay alone."*

*–(IDI, TB coordinator)*

Similarly, another health worker opined that;

*"Incentives like monetary allowances to treatment supporters can help motivate treatment supporters thus increase cure rates and reduce stigma to TB patients."*

*–(IDI, Nurse, Mma-Seetsele clinic)*

*Performance outputs*. **Service quality**. Interviewees expounded that service quality is a barrier to community TB care sustainability, with the majority reporting low DOT supervision by treatment supporters. The likelihood of poor adherence to DOT standards and procedures by treatment supporters was associated with a lack of training and incentives.
An interviewee pointed out;

*"Family members (Adherence buddy) tend to leave clients alone with medication without monitoring them hence clients defaulting treatment."*

*–(IDI, Nurse, Moshupa health post)*

This is corroborated by another interviewee who said;

*"I would explain to the patient how to take medication, because I would leave them with him to take when I'm not there because he knew the tablets."*

–(*SSI, Treatment supporter, Mma-Seetsele clinic*)

***Data reporting***. Data reporting also emerged as a barrier to community TB programs, with participants expressing irregularity and incompleteness of CHW reports on the services they provide at the community level.

A health worker stated that;

*"There is poor coordination between the doctor and the TB focal person as patients' information is mostly not comprehensively documented on patients' cards when the patient is initiated by the doctors e.g. contacts often not captured and the final treatment outcome is most of the time absent in the registries."*

–(*IDI, Data support officer, BUMMHI*)

This concurs with an observation that;

*"TB registers and patient facility cards had missing/incomplete data in almost all facilities eg monthly compliance with treatment, sputum smear results and treatment outcomes data missing."*

-(*Registers Observation,13<sup>th</sup> April 2022*)

***Job satisfaction and motivation***. Interviewees acknowledged reluctance to put forth and maintain effort on assigned TB activities, gaining less personal gratification from giving back to the community, providing services, and identifying a lack of motivation and reward as major hurdles to the program's execution.

An interviewee opined that;

*"I wish to be the change agent and improve TB patients' quality of lives in our community however being a treatment supporter is a lot of unpaid work, worse than that there is less support from family and health facility making work even more difficult therefore I don't think I can risk myself when I'm not happy."*

(*SSI, Treatment supporter, Moshupa health post*)

***Staff retention***. Staff attrition was identified as a potential obstacle to program implementation. Few participants addressed the fact that many practicing CHWs had resigned, retired, or given up their jobs, possibly due to a lack of motivation and incentives, consequently making treatment supporters inactive and negatively harming the program. An interviewee points out;

*"Treatment supporters are no longer active, with a smaller number of staff trained on TB case management transferred out of the DHMT, and resigning treatment supporter thus crippling the program."*

- (*IDI, Nurse, Moshupa SDA*)

***Community access***. Accessibility of TB services in the community has been cited as a hindrance to program implementation. Participants noted that clients did not often seek out or use promotional, preventative, or curative services, possibly due to a lack of knowledge.

An interviewee stated that.

*"CTBC uptake is very low mainly due to patients not being interested in the program and health care workers not educating and counselling clients thoroughly on the program."*

*–(IDI, TB coordinator)*

Other health workers submitted that;

*"Care givers are not co-operating with patients and health care workers as they often do not take medication well on time for their patients.)*

*- (IDI, TB screener, BUMMHI)*

*"There is poor community involvement as most clients(patients) and care givers are reluctant to enroll on community TB care."*

*–(IDI, Medical officer)*

***Credibility***. Implementing the program was hindered by patients' lack of trust in community health workers. As an interviewee opined that;

*"Some families felt we came to assess how they live or judge them when we go on family visit and that created some challenge in delivering the TB services"*

*- (IDI, TB screener, BUMMHI))*

One former red cross volunteer mentioned that patients preferred receiving care from nurses and doctors rather than from treatment supporters, because they had doubts or trust issues and believed that nurses are the ones who could give them quality, respectful care. As an interviewee points out;

*"Patients preferred nurses over us as the NGOs and doubted the quality of service we can provide."*

*–(FGD with Community leader)*

## Discussion

Community TB Programs are set up to improve the reach and sustainability of TB services and accelerate progress towards ending TB by 2030. The attainment of these goals depends on the availability of resources of the right kind, quantity, and mix. Ultimately, goal attainment depends on the proper utilization of these resources [14]. In this qualitative study, we explored factors supporting and hindering the implementation and sustainability of community-based interventions for the prevention and treatment of TB in Moshupa District, Botswana. We generated themes covering a variety of facilitators and barriers, stretching across all framework levels.

Our study findings highlighted effective partnerships between health services and external stakeholders as facilitators of community TB access by bringing services to people's homes and reducing the cost of care-seeking for patients and health services. These findings coincide with some studies reporting that, community and stakeholder involvement carefully designed community involvement initiatives facilitate community empowerment ultimately leading to improved health outcomes and sustaining the program [15–17].

A study by Zambia [18] documented that the presence of national policy and implementation documents at intermediate and operational levels is an important factor for successful

program implementation. These findings are consistent with our study that the presence of national and district policies and SOP's for Community TB care are facilitators and critical to providing strategic program direction, promoting good leadership, and governance for program implementation [18]. However, there is a need to strengthen policy support; for example, educational background was not included in treatment supporters' eligibility. This was supported by some studies stating that the depth and breadth of local and national policy documents, how and by whom they are formulated and prioritized, as well as their transferability and applicability to local contexts, should be examined and strengthened for sustainable TB control [19, 20].

The following challenges were noted: the program depended on the government health budget for TB program funding, which is often inadequate to fully manage the program activities. This, in turn, influences the availability of commodities (equipment and medicine), which are important drivers of direct medical costs and, thus, the need for sustainable funding. These findings are supported by a study indicating that adequate financing is needed to enable access, reduce delays, and compensate for direct and indirect costs, ultimately improving patient outcomes and program sustainability [21, 22]. Moreover, inadequate logistics has the potential to make a good intervention, such as community TB being misconstrued as non-performing, as it interrupts service delivery in the communities and affects the continuity of care, which hinders implementation and threatens communities 'trust and confidence in the program [18]. Therefore, to ensure sustained and effective program implementation, consistent district health system financing must be enhanced [18].

The lack of a community-based electronic system was found to be a barrier to community TB care, with the current paper-based system lacking organization for large data volumes, resulting in inconsistencies, incomplete data, poor data reporting, and typically lacking good tools for security and confidentiality. Contrary to the findings, studies in Angola [23] and Rwanda [24] have shown that, the integration of electronic-based reporting software decreases communication time, ensures consistency, and increases the completeness of reporting. suggesting that, presence of community based electronic systems improve buy-in by directly benefiting those collecting the data, unlike the paper-based one; they also warn doctors about important and urgent interventions resulting in improved patient outcomes and fewer required follow-up visits, thus strengthening program implementation [25, 26].

Our study revealed challenges in the lack of trained community health workers, which negatively influenced program implementation. These findings are consistent with a multi-country study in Afghanistan, Democratic Republic of Congo, Haiti, Somalia [27], and other studies in Malawi [28] and Tanzania [29] revealing hampered DOTS services expansion due to the limited capacity of trained staff, thus contributing to the high proportion of missing TB outcomes. Furthermore, the lack of adequately trained and qualified staff has been identified globally as a major factor preventing the successful achievement of TB targets. This lack of knowledge is likely to lead to extensive delays in treatment delivery experienced by patients [30]. These findings imply that service provider training is critical and should not be overlooked in efforts to enhance implementation outcomes, as CHWs are key drivers of the program.

While several factors may have contributed to poor community TB implementation, a lack of incentive was frequently noted as a barrier to community TB care implementation in Moshupa. The indirect costs associated with travelling, food, and accessories are often pronounced. These findings are in agreement with some studies in Zambia [18] Malawi [31, 32] and Nigeria [33] which reported a lack of incentive to affect motivation, retention of CHWs, and slow the rate of program implementation. Moreover, some systematic reviews found modest-to-moderate effects of financial incentives on healthcare worker practice [27] and

monetary incentives linked to motivation, performance, and attrition in community health worker programs [34]. Two systematic reviews, however, have found the impact of incentives to be less clear, noting the potential for important negative effects in neglecting unincentivized tasks [35, 36].

Insufficient monitoring, supervision of TB work, and motivation were the foremost factors identified to impede the successful functioning of the TB program in this study. These findings coincide with a study in Mozambique reporting that a lack of a motivated TB taskforce to supervise and monitor TB control activities in the health facility was a potential factor contributing to poor quality TB care [37]. Moreover, Mauch et al. emphasized the importance of program leadership and stewardship for quality and sustained TB control [27]. A study by Bennett et al. [38] in six sub-Saharan African countries also reported that leaders' weak ownership of the program negatively affects program coordination and implementation.

Community health worker trust by service beneficiaries was also reported as a barrier to program implementation and sustainability, with clients preferring nurses and doctors over treatment supporters. This mistrust could influence patient expectation satisfaction, thus determining the likelihood of future service utilization. This is consistent with a study from Nigeria [39] and a systematic review [40] demonstrating that community trust improves the quality of existing basic amenities, thus improving the acceptability of services and health-seeking behavior, ultimately contributing to TB program sustainability and improving population health [39, 40].

## Strengths and limitations of the study

We used experienced data collectors who were familiar with the Moshupa context, including community TB, which helped generate high-quality contextualized data for our study. In addition, triangulation of data and extensive discussion among members of the research team enabled us to better understand and ascertain the dependability, accuracy, breadth, and depth of the data collected. This study adds value to the existing knowledge by providing concrete insights to strengthen facilitators and overcome barriers related to community TB implementation.

The primary limitation of this study is its reliance on respondents' self-report data, which is subject to several sources of bias, including recall and social desirability biases. It is also important to recall that interviews are subjective in nature and contain the potential for misinterpretation of meaning through the process of translation and analysis. Some limitations of this study are selection bias and social desirability bias due to purposive sampling; however, researchers tried to mitigate social desirability bias by building rapport with the interviewees and by following statements made by interviewees with clarifying and probing questions during the interviews, rather than strictly adhering to the interview guides.

## Conclusion

This study provides insights into facilitators and barriers to the implementation and sustainability of community TB in Moshupa, as perceived by community health workers and stakeholders. The findings highlight inconsistencies in community TB care due to prevailing governance arrangements and supportive systems, which promote or hinder program implementation. The key facilitators and barriers highlighted should be considered by policymakers, district health managers, community TB supervisors, health facility managers, and district health co-operating partners in the design of context-specific health system strengthening interventions to ensure the effectiveness and sustainability of the community TB program implementation, thus accelerating progress to the End TB strategy targets. Additionally, a

comprehensive health promotion approach based on the Ottawa charter principles should be taken into account to curb barriers to community TB sustainability.

## Supporting information

**S1 File.**
(DOCX)

**S1 Table. Operational definitions of measurement constructs in the community health worker performance measurement framework [7].**
(DOCX)

**S2 Table. Sociodemographic attributes /characteristics of the respondents/participants from Moshupa, Botswana.**
(DOCX)

**S3 Table. Selected patient interview quotes categorized by the framework levels and barriers to and potential facilitators to the implementation and sustainability of the community-based tuberculosis care interventions.**
(DOCX)

## Acknowledgments

We are grateful to the Moshupa tribal headsman and the people of the study communities, as well as the management of the Moshupa district health management team for providing us with the necessary support during the data collection period upon which the findings of this study were based. We would also like to acknowledge Tshepo Keitshwaretse for his assistance during data collection.

## Declaration

**Consent for publication**. Written informed consent was obtained from all participants before their inclusion in the study. We also ensured the highest level of confidentiality and anonymity of the collected data.

## Author Contributions

**Conceptualization:** Gabalape Arnold Sejie, Ozayr H. Mahomed.

**Data curation:** Gabalape Arnold Sejie.

**Formal analysis:** Gabalape Arnold Sejie.

**Investigation:** Gabalape Arnold Sejie.

**Methodology:** Gabalape Arnold Sejie.

**Project administration:** Gabalape Arnold Sejie.

**Resources:** Gabalape Arnold Sejie.

**Supervision:** Ozayr H. Mahomed.

**Validation:** Gabalape Arnold Sejie.

**Visualization:** Gabalape Arnold Sejie.

**Writing – original draft:** Gabalape Arnold Sejie.

**Writing – review & editing:** Gabalape Arnold Sejie, Ozayr H. Mahomed.

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
