## [Decision Letter · Decision Letter 0]

16 Jan 2023

PONE-D-22-32443Potential facilitators and inhibitors to the implementation and sustainability of the community-based tuberculosis care interventions. A case study from Moshupa district, BotswanaPLOS ONE

Dear Dr. Sejie,

Thank you for submitting your manuscript to PLOS ONE. After careful consideration, we feel that it has merit but does not fully meet PLOS ONE’s publication criteria as it currently stands. Therefore, we invite you to submit a revised version of the manuscript that addresses the points raised during the review process.

We look forward to receiving your revised manuscript.

Kind regards,

Gifty Dufie Ampofo, M.D., Ph.D

Academic Editor

PLOS ONE

Journal Requirements:

6. We note that Figure S2 in your submission contain map image which may be copyrighted. All PLOS content is published under the Creative Commons Attribution License (CC BY 4.0), which means that the manuscript, images, and Supporting Information files will be freely available online, and any third party is permitted to access, download, copy, distribute, and use these materials in any way, even commercially, with proper attribution. For these reasons, we cannot publish previously copyrighted maps or satellite images created using proprietary data, such as Google software (Google Maps, Street View, and Earth). For more information, see our copyright guidelines: http://journals.plos.org/plosone/s/licenses-and-copyright.

a. You may seek permission from the original copyright holder of Figure S2 to publish the content specifically under the CC BY 4.0 license.  

Reviewers' comments:

Reviewer's Responses to Questions

**Comments to the Author**

1. Is the manuscript technically sound, and do the data support the conclusions?

Reviewer #1: No

Reviewer #2: Yes

2. Has the statistical analysis been performed appropriately and rigorously? 

Reviewer #1: No

Reviewer #2: Yes

3. Have the authors made all data underlying the findings in their manuscript fully available?

Reviewer #1: Yes

Reviewer #2: No

4. Is the manuscript presented in an intelligible fashion and written in standard English?

Reviewer #1: Yes

Reviewer #2: Yes

5. Review Comments to the Author

Reviewer #1: Reviewer comments

This paper is a potential contribution to the literature on community-based support for the management of tuberculosis in Botswana. It is based on an apparently failed community TB care (CTBC) program in Botswana. The study used a qualitative approach and all the elements of standard manuscript have been included. The quality of writing is acceptable although thorough proofreading will be needed before publication.

Major concern

•The description of the methods section is not detailed enough to permit replication of the study without recourse to the authors. The design is described in a maze of imprecise lexicon –“Interpretive 7 paradigm based on relativist ontology and subjectivist epistemology along with adductive research”. This should be revised. As nebulous are these are, there is very little reflection of their application in the analysis and reporting of the study.

•Important details like who conducted the interviews, where they were conducted, the nature of interview guide, how was the FGD organized? Was it one FGD with 8 community leaders?

•It is reported that clinic-based observations were conducted. There is however no information on how they were conducted. By whom? When and how? Was it with a standard tool? How was analysis of that data conducted?

Other concerns

•The descriptive results in the opening paragraph of results is not comprehensive enough. The authors need to detail out the overall picture before going into the description by gender and age. And for each of the approaches to data collection, the authors need to be detailed in the characteristics of the participants

•The results should also present a triangulation of information and not just a topic by topic representation with quotes. It is out of the triangulation of information that new understanding or theory emerges to add something new to the literature.

•While it is well to present the Botswana program as having failed and needing to be researched, it is important for evidence to that effect to be presented. The authors need to provide data to back the claim of “low treatment outcome (this statement is unclear)” and “high case notification”? There should also be an attempt to link the above shortcomings to the delivery of Community TB Care.

Reviewer #2: REVIEW OF “POTENTIAL FACILITATORS AND INHIBITORS TO THE IMPLEMENTATION AND SUSTAINABILITY OF THE COMMUNITY-BASED TUBERCULOSIS CARE INTERVENTIONS. A CASE STUDY FROM MOSHUPA DISTRICT, BOTSWANA”

This qualitative study sought to understand barriers and facilitators to TB treatment in a community Botswana, which is an important study for the sub-Saharan African region.

General comments

•Is Moshupa a community, district? This is very confusing.

•Appropriate or standard referencing style should be used, authors place the references after full stops.

oLine 371 and in several other sentences authors need to how they apply the references

oLine 380 and several others: authors need to check the closeness of reference to words, no space etc.

•The entire manuscript needs some editing to improve the language quality. The following are only but a few examples:

o Sometimes articles are missing, singular and plural tenses are not respected eg (line 398 “lack of incentives was frequently noted barriers to”

oSome of the sentences need rephrasing eg line 400” “The indirect cost associated with the traveling, food and accessories were often pronounced.”; lines 400-402: … in Zambia(17) Malawi(31)(32) and Nigeria(33) which reported lack of incentives to affect motivation, retention of CHWs and slows the rate of program implementation.

•Very little literature has been explored in the introduction and discussion sections

Abstract

•Check the use of tenses such as singular and plural in the abstract, eg “Initiative employees for in-depth interview and 8 community leaders for focus group discussions”

•“the HIV negatives and 62/100 000 among the HIV positive”

•Check parallel structure in sentences eg. Line 52 “decentralizing TB care beyond health facilities and harness the”

•Check the use of capitals and small letters eg: Line 53 “The Botswana National Tuberculosis and leprosy Program (BNTP)”

Introduction

•Authors should provide a brief description of the current state of TB policy or guidelines for implementation in the introduction. Later in the results, they can use real data to explain how the policy is actually being implemented, if this was part of their objective for the study.

•There is no conceptual framework yet. The authors need to present a detailed well explained conceptual framework. The framework should explain the key concepts and how they were applied in the study.

Methods

•Setting: No justification for the selection of Moshupa village.

Study design and population

•Justification for each group of participants selected is missing

•What kind of information did each category of IDI/FGD guide seek to solicit? How many categories of IDIs/FGDs were developed and for which categories?

•What do authors consider as groups?

•Line 104: What does "self developed mean" mean?

Procedures

•Who collected the data?

Data analysis

•Nvivo version?

•How were texts coded? Manual or with the support of Nvivo?

•Were the notes also transferred to Nvivo for coding and analyses?

•Who analyzed the data?

•Kindly explain further the following statement: “Quality was ensured through joint assessment of the sampling approach and ongoing review of transcripts to explore areas for further probing"

Results

Sociodemographic attributes of the respondents

•This is supposed to be a purely qualitative paper. Lines 85-90 in the methodology section report on very diverse groups of respondents who play different roles and have different experiences. How were these diverse groups lumped into one category for statistical analysis, what is the implication for the study itself and the results presented?

•This study should remain a qualitative study, authors should not attempt to confuse the study. Thus, authors should take out the statistics and present a write up and table displaying the different categories and the background characteristics that is relevant to this study such as age, education, occupation/designation among others.

Qualitative findings

•Line 139: Take off “Qualitative” out of findings, the entire study is quantitative.

•What does line 146 “input” mean? This is incomplete.

•Line 143: authors should present a table on the main theme and sub themes in the manuscript and if they still want to maintain S3 Table, they can do so

•Line 148, which researchers are you referring to?

•Under introduction to findings lines 140-143, kindly give a summary of the findings that authors are going to present in detail in the subsequent write up.

•Under Policies and Standard Operating Procedures: Authors are not presenting findings of the study, but their own conjecture. This section should be expunged. Authors should explain what the policies and guidelines are in the introduction section, so that they can later report on how they are being implemented in the results section based on the data they collected and NOT from the RESEARCHERS’ perspective.

•Findings from lines 161 to 339, do not read like qualitative results but a listing of items. I am afraid, but the authors will need to do a more in depth analysis of the data to present actual findings that show the nuances, the depth of the issues studied, the themes among others. I strongly recommend that the authors should do a second level analysis of the critical or interesting issues report on. Also, authors interviewed different categories of respondents, but the different perspectives of these respondents are not reported in the different sections. The reader needs to read the similarities, contradictions among others from the different perspectives of the diverse groups of respondents. The sub headings are numerous and each contains very scanty information. I recommend that the authors should identify critical themes based on the objectives of the and do a thorough narration.

•Sometimes the reader gets confused between a quote, a reference to policy document etc. I recommend that authors do proper quoting such as (a). indent quotes from the write up, (b). include quotation marks

•Authors present quotations and attributes, without indicating the event such as whether it was in an IDI or a focus group discussion. In presenting observations, authors should do well to indicate where the observation took place, such as in a TB clinic, office etc and also include the date of the observation.

Discussion

•Lines 347- 348: What does the following mean? “…… which we divided into various categories stretching across all framework levels.” Were the divisions made out of convenience or based on themes generated from the analysis of data? This should be clearly stated.

•How do lines 380-382 support lines 377-380, considering that they are opposites, I thought that they contrast rather?

•380: why do authors state that “this was supported” instead of “this is supported by”. Also, what does “This” mean?

•Line 390: what is DR Congo?

•Line 386: what does ‘human resources’ mean? Be specific.

•Lines 389-391: is this one study or several studies? The narration appears to be two different studies.

•Line 398: several factors contributed to what?

•In some of the paragraphs the authors refer to literature firs eg lines386-388. Besides the first paragraph that presents a summary of the findings the first sentence in the subsequent paragraphs should report on a finding first before presenting literature to compare, contrast among others.

Strength and Limitation

•The heading Strength and Limitation” is incomplete and should be in the plural

•Line 436: Authors state: “Some limitations of this study are selection and social”, please elaborate on them

•Authors report triangulating data and analysis as a strength, yet I did not find triangulation reported in data analysis.

Conclusion

Authors should present two to three strong recommendations based on their findings in the concluding section

Consent for publication

•How was confidentiality and anonymity in data collection ensured?

6. PLOS authors have the option to publish the peer review history of their article (what does this mean?). If published, this will include your full peer review and any attached files.

Reviewer #1: No

Reviewer #2: No

---

## [Author Response · Author response to Decision Letter 0]

12 Apr 2023

Kindly receive the corrected manuscript and thank you for patience and support, much appreciated. Looking forward to publication with your prestigious journal

I have attached a revised manuscript together with responses to reviewers

---

## [Decision Letter · Decision Letter 1]

22 Jun 2023

PONE-D-22-32443R1Potential facilitators and inhibitors to the implementation and sustainability of the community-based tuberculosis care interventions. A case study from Moshupa, BotswanaPLOS ONE

Dear Dr. Sejie,

Thank you for submitting your manuscript to PLOS ONE. After careful consideration, we feel that it has merit but does not fully meet PLOS ONE’s publication criteria as it currently stands. Therefore, we invite you to submit a revised version of the manuscript that addresses the points raised during the review process.

Kindly consider the second reviewer's comments for minor revision, especially in formatting the quotes and giving some interpretation to the findings for a final manuscript to be considered for publication.

We look forward to receiving your revised manuscript.

Kind regards,

Gifty Dufie Ampofo, M.D., Ph.D

Academic Editor

PLOS ONE

Journal Requirements:

Reviewers' comments:

Reviewer's Responses to Questions

**Comments to the Author**

1. If the authors have adequately addressed your comments raised in a previous round of review and you feel that this manuscript is now acceptable for publication, you may indicate that here to bypass the “Comments to the Author” section, enter your conflict of interest statement in the “Confidential to Editor” section, and submit your "Accept" recommendation.

Reviewer #1: All comments have been addressed

Reviewer #2: All comments have been addressed

2. Is the manuscript technically sound, and do the data support the conclusions?

Reviewer #1: Yes

Reviewer #2: Yes

3. Has the statistical analysis been performed appropriately and rigorously? 

Reviewer #1: Yes

Reviewer #2: N/A

4. Have the authors made all data underlying the findings in their manuscript fully available?

Reviewer #1: Yes

Reviewer #2: No

5. Is the manuscript presented in an intelligible fashion and written in standard English?

Reviewer #1: Yes

Reviewer #2: Yes

6. Review Comments to the Author

Reviewer #1: I have reviewed the paper and I am satisfied that the authors have addressed all the concerns pointed out by the reviewers. The paper is substantially improved. I do not have any reason to be concerned about dual publication, research ethics, or publication ethics

Reviewer #2: Second review of “Potential facilitators and inhibitors to the implementation and sustainability of the community-based tuberculosis care interventions. A case study from Moshupa, Botswana”

• The authors need to edit the entire manuscript for wrong tenses and missing articles especially the sections in track changes

• Page 18 line 219 has an incomplete sentence

• The quotations are not aligned, they are not indented and they do not have quotation marks. Also, the authors do not use the appropriate formatting for quotations. I recommend that they refer to the AAA style guide to carry out the necessary revisions (American Anthropological Association. AAA Style Guide 2009, 2009. Available from: https://socioanthro.uoguelph.ca/sites/default/files/page-files/American-Anthropological-Association-Style-Guide.pdf.).

• The presentation of the findings is very poorly done, the authors do not make efforts to interpret the results before citing the quotations. I urge the authors to read a few well written qualitative articles to improve upon the presentation of the results.

• The in text citations within the information in tracks are not are not properly done.

• The tables within the manuscript are labeled as S1 Table etcetera?

7. PLOS authors have the option to publish the peer review history of their article (what does this mean?). If published, this will include your full peer review and any attached files.

Reviewer #1: No

Reviewer #2: No

---

## [Author Response · Author response to Decision Letter 1]

28 Jul 2023

1. The authors need to edit the entire manuscript for wrong tenses and missing articles especially the sections in track changes. Response: Tenses corrected see yellow coloured words and tracked changes on revised paper

2. Page 18 line 219 has an incomplete sentence. Response: Sentence completed,see Page 20

3. The quotations are not aligned, they are not indented, and they do not have quotation marks. Also, the authors do not use the appropriate formatting for quotations. I recommend that they refer to the AAA style guide to carry out the necessary revisions (American Anthropological Association. AAA Style Guide 2009, 2009. Available from: https://socioanthro.uoguelph.ca/sites/default/files/page-files/American-Anthropological-Association-Style-Guide.pdf.). Response: Quotations inserted and quotes indented, see Page 19-31

4. The presentation of the findings is very poorly done, the authors do not make efforts to interpret the results before citing the quotations. I urge the authors to read a few well written qualitative articles to improve upon the presentation of the results. Response: Results interpretation addressed Page 19-31

5. The in-text citations within the information in tracks are not not properly done. Response: Change incorporated 

6. The tables within the manuscript are labelled as S1 Table etcetera? Response: The journal Submission Guidelines requires they be labelled that way.

---

## [Editor Report · Decision Letter 2]

1 Aug 2023

Potential facilitators and inhibitors to the implementation and sustainability of the community-based tuberculosis care interventions. A case study from Moshupa, Botswana

PONE-D-22-32443R2

Dear Dr. Sejie,

We’re pleased to inform you that your manuscript has been judged scientifically suitable for publication and will be formally accepted for publication once it meets all outstanding technical requirements.

Kind regards,

Gifty Dufie Ampofo, M.D., Ph.D

Academic Editor

PLOS ONE
---

## [Editor Report · Acceptance letter]

3 Aug 2023

PONE-D-22-32443R2 

Potential facilitators and inhibitors to the implementation and sustainability of the community-based tuberculosis care interventions. A case study from Moshupa, Botswana 

Dear Dr. Sejie:

I'm pleased to inform you that your manuscript has been deemed suitable for publication in PLOS ONE. Congratulations! Your manuscript is now with our production department. 

Kind regards, 

on behalf of

Dr. Gifty Dufie Ampofo 

Academic Editor

PLOS ONE